# Health-Related Quality of Life and Family Functioning of Primary Caregivers of Children with Menkes Disease

**DOI:** 10.3390/jcm12051769

**Published:** 2023-02-22

**Authors:** Anna Rozensztrauch, Izabela Dzien, Robert Śmigiel

**Affiliations:** Department of Nursing and Obstetrics, Division of Family and Pediatric Nursing, Wroclaw Medical University, 50-367 Wroclaw, Poland

**Keywords:** Menkes disease, quality of life, family impact, rare disease

## Abstract

Background: Menkes disease (MD; OMIM #309400) is a progressive neurodegeneration that results from abnormalities in the copper metabolism which are already present before birth. It is an extremely rare condition. The study was conducted to assess the quality of life of children with MD syndrome and the impact of the disease on family functioning. Methods: A cross-sectional questionnaire survey was used. The subjects were 16 parents of children with MD. The method used was the Paediatric Quality of Life Inventory and the PedsQL Family Impact Module and the author’s own questionnaire. Results: Quality of life (QOL) was 29.14 (SD = 14.73), with the lowest for physical functioning (M = 10.55; SD = 10.26) and highest for emotional functioning (M = 48.13; SD = 29.43). The highest score was on the family relationships domain (M = 56.25, SD = 20.38) and the cognitive functioning domain (M = 50.00, SD = 19.24) and the lowest was on the daily activities’ domain (M = 32.29, SD = 20.38) and the physical functioning domain (M = 39.84, SD = 14.90). The analysis did not show statistically significant relationships between age (*p* = 0.193) and the number of epileptic seizures a week (*p* = 0.641) and the overall QOL of the children studied. No statistically significant relationships were found between treatment with copper histidine and the overall QOL of the children (*p* = 0.914) and in physical functioning (*p* = 0.927), emotional functioning (*p* = 0.706), and social functioning (*p* = 0.751). The presence of comorbidities did not have an influence on the overall QOL. Conclusions: MD has a moderate impact on the functioning of the families of the affected children. The age of the child, number of epileptic seizures a week, feeding method (oral feeding or feeding via a PEG tube), and treatment with copper histidine do not have a significant impact on the QOL of children with MD.

## 1. Introduction

Menkes disease (MD; OMIM #309400), which is also referred to as Menkes kinky hair disease because of the characteristic appearance and structure of the hair of affected children, is a rare neurodegenerative disorder. In MD, progressive neurodegeneration results from abnormalities in the copper metabolism which are already present before birth. Classical MD is characterised by a severe phenotype and usually leads to death in early childhood [1,2].

It is an extremely rare condition; its incidence in Europe is estimated at 1/300,000 live births. The incidence of MD in Australia is higher than elsewhere, which might be due to the founder effect [3,4].

MD is a genetic X-linked recessive disorder. It is caused by mutations in the ATP7A gene, which is located on the X chromosome (xq21.1) and encodes a copper-transporting P-type ATPase. A number of different mutations in the ATP7A gene have been reported in children with MD (including insertions, deletions, and splice-site mutations). However, no significant correlation exists between the mutations and the clinical course of the disease [5]. MD mainly affects boys; however, several cases of female MD patients have been reported worldwide [3,5,6].

The ATP7A gene encodes a copper-transporting protein. The impaired function of the gene results in abnormal distribution of copper throughout the body; it accumulates in excessive amounts in tissues, such as the kidneys, and is significantly deficient in the brain, liver, and blood. A deficiency or impaired activity of copper-dependent enzymes causes the symptoms of the disease [7].

The symptoms of MD are a direct consequence of the dysfunction of copper-dependent enzymes or are secondary to the inability to load the enzymes with copper [8]. For example, the deficiency of cytochrome oxidase impairs cellular respiration, which results in the degeneration of the central nervous system and ataxia, whereas hypotension and hypothermia are due to abnormal catecholamine synthesis.

Pregnancy is usually uncomplicated. The possible clinical features of MD in the neonatal period include prolonged jaundice, hypothermia, hypoglycaemia, and feeding problems [2]. The first sign of MD may be kinky, dull, steel, and wool-like hair. MD patients’ hair is sparse, short, and hypopigmented and its microscopic examination reveals four characteristic findings: pili torti, which is 180° twisting of the hair shaft, trichorrhexis nodosa, which is transverse fractures of the hair shaft at regular intervals, trichoptilosis, which is longitudinal splitting of the hair shaft, and monilethrix, which is where hair shafts are of a varying diameter [9].

Infants with MD also have dysmorphic facial features such as frontal or occipital bossing, pudgy cheeks, a broad nasal bridge, and micrognathia [2]. Other clinical features include pale, hypopigmented skin [9,10] as well as an amimic face and drooping eyelids.

Children with MD usually develop normally until the age of 2–4 months [6]. Then, they develop epileptic seizures, stop learning new skills, and show regression of developmental milestones [2]. In the first months of life, hypotonia is present, which later progresses to spasticity and paresis of the limbs [6,10].

Epileptic seizures usually appear at the age of 6–8 weeks. Seizure types and EEG characteristics evolve over time [10]. Patients usually initially experience focal seizures, which then progress to status epilepticus (<3 months) and epileptic spasms (6–11 months). At a later stage, multifocal seizures, tonic spasms, and myoclonus develop [11,12].

The most common urological problem in patients with MD are bladder diverticula. Other possible urological symptoms include neurogenic bladder and bladder wall thickening. All these urological problems may predispose to recurrent urinary tract infections [13].

The diagnosis of MD is based on biochemical findings (low serum copper and ceruloplasmin levels) and clinical features [14].

A definitive diagnosis of MD can only be made by molecular genetic testing. After DNA is isolated from the material collected (most commonly blood), it is tested for mutations. Patients with MD carry mutations in the ATP7A gene, which is located on the Xq13.3 chromosomal region. Due to the large size of the gene and the variety of mutations that can occur in the gene, it may take time to detect the genetic defect responsible for the condition in a given family [6]. As approximately one-third of ATP7A mutations are exon deletions, multiplex ligation-dependent probe amplification (MLPA), a PCR-based technique, can be performed as the first test. Small intragenic ATP7A mutations can be detected by PCR amplification and the sequencing of all coding exons using genomic DNA. Deletions can also be identified by genomic sequencing [10].

There is a biochemical test that definitively establishes a diagnosis of MD. It is based on the assessment of intracellular copper accumulation due to an impaired efflux. The accumulation of copper is evaluated in cultured cells, usually fibroblasts. However, such analyses are carried out in only a few specialised centres around the world [6].

The early diagnosis of MD is challenging as the clinical presentation and biochemical markers may initially be nonspecific. Serum copper and ceruloplasmin levels can also be low in healthy newborns [2]. In the neonatal period, plasma catecholamine analysis indicative of dopamine β-hydroxylase deficiency may be used as a diagnostic test for suspected MD. However, the test does not establish a definitive diagnosis of MD. Diagnostic imaging can, too, be helpful in the diagnosis of MD. The tortuosity of cerebral blood vessels and demyelinating changes in the white matter are the main brain MRI findings in MD patients [9]. As the disease progresses, cerebral and cerebellar atrophy develops. Other possible brain MRI findings include focal lesions involving basal ganglia and subdural collections [15].

While prenatal diagnosis is available, it is only performed if there is an affected patient or a known ATP7A mutation in the family. If the ATP7A mutation in the family is known, it can be investigated in the DNA isolated from chorionic villi or amniotic fluid as part of a prenatal diagnosis. In high-risk pregnancies, where the mutation in the family is not known, biochemical analysis can be performed as it may be challenging to identify the genetic defect in a limited timeframe. In the first trimester, the total copper content in chorionic villi can be determined directly by neutron activation analysis or atomic absorption. Between weeks 14 and 27 of pregnancy, the accumulation of copper is evaluated in cultured amniotic fluid cells. Such analyses have been routinely performed at the Kennedy Centre in Denmark since 1975 [6].

There are currently no newborn screening programmes for MD. While abnormal plasma neurochemical levels due to dopamine-β-hydroxylase deficiency are diagnostic in newborns with MD, this approach is impractical for newborn dried blood spot screening due to the need for a larger blood volume as well as an alumina extraction step [16].

Children with the classical, severe form of MD usually die before the age of 3. However, careful care by an extensive medical team and the patient’s family, and possibly a copper administration, may extend the life span of MD patients to 13 years or for even longer [6]. Because of the severe, progressive nature of the disease, many patients with MD are in hospice care.

The available treatment option consists of the administration of copper histidine. It is recommended that it is administered in the form of daily subcutaneous injections. The oral administration of copper histidine is not effective as copper is not absorbed from the gastrointestinal tract [6]. The dosage of copper histidine is determined individually for each patient and ranges from 200 to 1000 µg per day [17]. The response to treatment depends on the degree of ATP7A mutation. If some capacity for copper transport is retained, the prospects for successful neurological outcomes are higher [18]. It is very important that treatment is started as soon as possible after birth, preferably before symptoms start. It can be initiated as early as the prenatal period [19].

MD is an incurable disease that affects the functioning of various body systems. Therefore, patients with MD require holistic care by their family and a multidisciplinary medical team. Neonatal medical personnel, including nurses, midwives, and neonatologists, play an important role in that they can recognise such early signs of MD as unusual hair or dysmorphic facial features [2].

The developmental delay and epileptic seizures suffered by children with MD prompt an early referral to a neurologist. It is neurologists who often refer their patients to genetic or metabolic specialists. Geneticists provide the parents of children with MD with reproductive risk and prenatal diagnosis counselling. As MD patients have feeding difficulties and often fail to gain the expected weight, most of them are referred to a feeding clinic and a dietician [2].

The study was conducted to assess the quality of life of children with MD syndrome and the impact of the disease on family functioning.

## 2. Material and Methods

The study was carried out among 16 children with MD, including 6 children from Poland (one of them lived in the UK with their family), 4 children from the USA, 2 children from Italy, 1 child from India, 1 child from Spain, 1 child from the Netherlands, and 1 child from Sweden. All the children were male. The youngest child was 5 months old and the oldest was 15 years old. Nearly half of the children were under the age of 3.

The inclusion criteria were as follows: diagnosis of MD confirmed by molecular genetic testing and being aged under 18.

The exclusion criteria were as follows: death of the child, age over 18, failure to complete all study questionnaires, and suspected MD (failure to confirm diagnosis by genetic testing).

The study was carried out using standardised questionnaires assessing children’s health-related quality of life (HRQOL), the impact of the child’s disease on family functioning and the parents/caregivers’ perceived social support, as well as using the author’s own questionnaire.

The study was performed by means of a diagnostic survey using questionnaires. The following research tools were used in the study:

### 2.1. The Author’s Own Questionnaire

The author’s own questionnaire included questions about sociodemographic characteristics (age, sex, place of residence, whether there is more than one child in the family), clinical data (age at which first symptoms appeared, age at which the diagnosis of MD was confirmed, symptoms present in the child, feeding method, whether the child needs a urinary catheter, average number of epileptic seizures a week), the perceived burden of caring for a child with a chronic condition, impact of the child’s disease on parents’ social and work life (working status, participation in and hosting of social gatherings and family parties, pursuit of interests), and perceived social support (whether the child is in hospice care, whether the parents have someone in their close circle who can take care of their child on their own, and whether they have to employ the services of a nurse). The parents participating in the study were also asked whether, in their opinion, their child’s illness had negatively affected the quality of life of their other children (if there is more than one child in the family).

### 2.2. Generic Health-Related Quality of Life Questionnaire

The assessment of the QOL questionnaires was conducted with the use of a validated generic *PedsQL* questionnaire. This was the *Paediatric Quality of Life (PedsQL) 4.0 questionnaire*, which is composed of 23 items and assesses QOL within the preceding month in pre-school children (aged 5–7 years), school children (aged 8–12 years), and adolescents (aged 13–18 years). The report for children aged 2–4 years comprises 21 items and does not include school functioning and communication scales. Respondents rate the items on a 5-point Likert scale, where 0 denotes “never” and 4 “almost always”. The questionnaire items are reverse scored and linearly transformed to a scale of 0–100 (0 = 100, 1 = 75, 2 = 50, 3 = 25, 4 = 0), with 100 denoting the best QOL. The total score is the sum of the mean scores from each of the subscales. The lower the score, the lower the quality of life [20,21,22,23].

### 2.3. The PedsQL 2.0 Family Impact Questionnaire

The PedsQL 2.0 Family Impact Questionnaire contains 8 modules with a total of 36 sub-items for parents to rate on a scale from 0 (never) to 4 (almost always). The number of points assigned to a module will determine in which zone the family of a child struggling with MD disease has the most problems [24].

### 2.4. The Multidimensional Scale of Perceived Social Support (MSPSS)

The Multidimensional Scale of Perceived Social Support (MSPSS) by Zimet et al., as adapted into Polish by Buszman and Przybyła-Basista, consists of 12 items that measure perceived social support from family, friends, and significant others. The parents included in the study were asked to rate how much they agree or disagree with each item using a 7-point Likert scale, where 1 = “very strongly disagree” and 7 = “very strongly agree” [25].

The study was approved by the Bioethics Committee of the Wrocław Medical University, approval no. 130/2022.

## 3. Statistical Analysis

In order to assess whether, and to what extent, caring for a child with MD is burdensome for the child’s parents, the following questions were included in the author’s own questionnaire:-Do you feel burdened by caring for your child?-Do you have the time to pursue your hobbies/interests?-Do you feel socially isolated because of caring for your child?-Is there anyone (without medical training) in your close circle who can take care of your child on their own while you are away?-Do you employ the services of a nurse to care for your child?-Do you take part in social gatherings/family parties?-Do you hold social gatherings/family parties at home?-Do you have contact (face to face or via the Internet or telephone) with other parents of children with MD?-Do you think you have adequate knowledge about MD and about caring for a child with the condition?

In order to verify the research hypotheses formulated, statistical analyses were performed using the IBM SPSS Statistics 26 software package. The software was used to carry out an analysis of basic descriptive statistics, including the Shapiro–Wilk test, Pearson’s r correlation analysis, Spearman’s rho correlation analysis, and the *t*-test for independent samples. The level of significance was set at α = 0.05.

In the first step of the analysis, the distribution of quantitative variables was examined in order to assess the parents’ perceived burden of caring for a child with MD and their perceived social support as well as the functioning of children with MD and their families. To this end, basic descriptive statistics were calculated, including the Shapiro–Wilk test for normality.

The next step of the analysis was to assess whether the child’s age, number of epileptic seizures a week, feeding method, and treatment with copper histidine have a relationship with the HRQOL of children with MD. To this end, Pearson’s r correlation analysis was performed for independent quantitative variables, Spearman’s rho correlation analysis was carried out for independent ordinal variables, and Student’s *t*-test for independent samples was performed for independent nominal variables.

It was then determined whether there was a relationship between the feeding method and treatment with copper histidine and the HRQOL of the MD patients studied.

The final step of the analysis was to assess whether the duration of disease and the presence of comorbidities affect the HRQOL of children with MD and the functioning of their families. To this end, as in the previous case, Pearson’s r correlation analysis was performed for independent quantitative variables and Student’s *t*-test for independent samples was performed for independent nominal variables.

First, the relationship between the duration of disease and the HRQOL of the children studied and the functioning of their families was assessed. It was then assessed whether the presence of comorbidities affects the HRQOL of the MD patients studied and the functioning of their families.

## 4. Results

The children included in the study were aged between 5 months and 15 years. Almost half of the children (44%) were aged under 3, 19% were between 3 and 6 years old, 6% were between 6 and 10 years old, and 31% were over 10 years old.

Half of the respondents lived in cities of more than 100 thousand inhabitants. A slightly smaller percentage of respondents (44%) lived in rural areas. Over half of the parents who completed the study questionnaires (*n* = 10) were employed. In almost 70% of the families included in the study, 1 of the parents gave up working after their child’s diagnosis of MD. Ten of the children included in the study (*n* = 10) had siblings; one of the mothers participating in the study was pregnant. The majority (70%) of parents with more than one child stated that they were not able to pay proper attention to their healthy children. Two respondents reported that MD had been previously diagnosed in their family. One-fourth of the children studied (*n* = 4) were born by caesarean section. The remaining children (*n* = 12) were born vaginally. Slightly more than half of the children studied (56%) were born at term (between 37 and 42 weeks’ gestation). In total, 7 children (*n* = 7) were born preterm between 32 and 36 weeks’ gestation. Half of the children studied (*n* = 8) were in hospice care. Three quarters of the children included in the study (*n* = 12) did not need a urinary catheter. Three children (*n* = 3) required periodic catheterisation and one child (*n* = 1) required intermittent catheterisation (several times per day).

In order to carry out statistical analysis of the parents’ perceived caregiver burden, the answers provided by the parents to the questions listed in the description of statistical analysis were assigned scores. The analysis showed that the parents of MD patients included in the study had a moderate caregiver burden (M = 4.88, SD = 1.93). One parent (*n* = 1) reported that they did not find caring for their sick child burdensome at all. The vast majority (69%) of the parents stated directly that they felt burdened by caring for their child. Thirteen (*n* = 13) of the parents reported that they did not have anyone (without medical training) in their close circle who could take care of their child on their own while they are away. All the respondents (*n* = 16) reported that they have contact with other families of children with MD. Only 13% of respondents reported that, in their opinion, their knowledge of MD is insufficient. The analysis showed that the parents included in the study had high perceived social support (M = 66.94, SD = 12.47). The results of the Shapiro–Wilk test for variables concerning the perceived caregiver burden and the perceived social support of parents of MD patients were statistically insignificant. The results of the analysis are shown in Table 1.

An analysis of the HRQOL of the MD patients studied showed that their mean HRQOL rating was 29.14 (SD = 14.73). The parent-reported HRQOL scores of the patients were lowest for physical functioning (M = 10.55; SD = 10.26) and highest for emotional functioning (M = 48.13; SD = 29.43). The data are shown in Table 2.

In the case of most of the variables analysed, the Shapiro–Wilk test result was statistically insignificant, which means that their distribution was not significantly different from normal distribution. One exception was the variable concerning the physical functioning of the child, whose distribution was positively skewed. This means that in the case of the group studied, there was a larger number of low values of the variable.

Table 3 indicates a comparison of PedsQL score domain with the control group [20].

The mean overall rating for the impact of MD on family functioning was 44.16 (SD = 17.40). The parents included in the study scored highest on the family relationships domain (M = 56.25, SD = 20.38) and the cognitive functioning domain (M = 50.00, SD = 19.24) and lowest on the daily activities domain (M = 32.29, SD = 20.38) and the physical functioning domain (M = 39.84, SD = 14.90). The results of the analysis, including the results of the Shapiro–Wilk test, are shown in Table 4.

The parents participating in the study were asked about the average number of epileptic seizures their children experience a week. In total, 6 children (*n* = 6) had over 30 epileptic seizures a week, despite antiepileptic treatment; 31% of the children studied had between 1 and 10 seizures a week; and 3 children (*n* = 3) did not suffer from epileptic seizures.

The analysis did not show statistically significant relationships between age (*p* = 0.193) and the number of epileptic seizures a week (*p* = 0.641) and the overall HRQOL of the children studied. Moreover, no statistically significant relationships were demonstrated between age and the number of seizures a week and particular domains of HRQOL of the children studied, i.e., physical functioning (age *p* = 0.519, seizures *p* = 0.891), emotional functioning (age *p* = 0.315, seizures *p* = 0.663), and social functioning (age *p* = 0.268, seizures *p* = 0.801). This means that the parent-reported HRQOL scores of the children studied were similar regardless of the age of the children and the number of seizures they experience a week. The data are shown in Table 5.

When assessing the relationship between the feeding method and the HRQOL of the children studied, only those children who were fed orally or who had a percutaneous endoscopic gastrostomy (PEG) in place were included in the analysis as only two children (*n* = 2) were fed through a gastric tube. Of the children included in the analysis, seven (*n* = 7) were fed orally and seven (*n* = 7) were fed through a PEG tube.

Of the sixteen children studied (*n* = 16), only four (*n* = 4) were receiving treatment with copper histidine. Almost one-third (*n* = 5, 31%) of the children included in the study had previously been treated with copper histidine, but their parents decided to discontinue the treatment. The parents of five children (*n* = 5) decided not to pursue the treatment despite having been informed by doctors that it was available. Two parents (13% of the respondents) reported that they had not been offered treatment with copper histidine. Both families were from the USA.

The analysis did not demonstrate statistically significant relationships between the feeding method and the overall HRQOL of the children studied (*p* = 0.222) and their HRQOL in physical functioning (*p* = 0.642), emotional functioning (*p* = 0.622), and social functioning (*p* = 0.074). No statistically significant relationships were found between treatment with copper histidine and the overall HRQOL of the children studied (*p* = 0.914) and their HRQOL in physical functioning (*p* = 0.927), emotional functioning (*p* = 0.706), and social functioning (*p* = 0.751). The data are shown in Table 6.

The author’s own questionnaire included questions about the age at which the symptoms of MD first appeared and about the age at which the disease was confirmed by genetic testing. For the purpose of the study, the duration of disease was calculated from the onset of the first symptom. In 50% of the children studied, the first symptoms appeared at the age of 2–3 months, which is consistent with the data reported in the literature on MD. In one child (*n* = 1), the first signs of MD appeared in the prenatal period.

The analysis did not demonstrate any statistically significant relationships between the duration of disease and the HRQOL of the children studied (*p* = 0.199) and the functioning of their families (*p* = 0.560). This means that the mean scores for the HRQOL of the children studied and the functioning of their families, both overall and across particular domains, were similar regardless of how long the children and their families had been dealing with MD. The detailed results of the analysis are shown in Table 7.

Half of the children studied (*n* = 8) had comorbidities. Twenty-five per cent of the MD patients included in the study had epilepsy. Other comorbidities reported for the patients were respiratory diseases (frequent pneumonias) and urinary tract diseases. One child (*n* = 1) had Dandy–Walker syndrome. One child (*n* = 1) had another rare inherited disorder, namely, aminoacylase 1 deficiency.

No statistically significant relationship was demonstrated between the presence of comorbidities and the HRQOL of the children studied and the functioning of their families. One exception was HRQOL in emotional functioning (*p* = 0.028): children without comorbidities had a significantly better emotional functioning compared to children with comorbidities.

No statistically significant differences were found for other variables, which means that the presence of comorbidities did not have an influence on the overall HRQOL of the children studied, their HRQOL in physical and social functioning, and the functioning of their families. However, it can be noted that the presence of comorbidities had an influence on the mean total HRQOL scores and mean scores for the social functioning of the children studied. Children with comorbidities and those without comorbidities had the same mean HRQOL scores for physical functioning (M = 10.55). Parents of children with comorbidities reported slightly lower scores for the family relationships domain compared with the parents of children without comorbidities. However, they reported higher mean family functioning summary scores and higher mean scores for the daily activities domain compared with the parents of MD patients without comorbidities. Detailed results are shown in Table 8.

## 5. Discussion

To the best of the author’s knowledge, the present study is the first to investigate the HRQOL of children with MD and the impact of the disease on family functioning. As no other studies on this issue have been published to date, the findings from the present study cannot be compared with those from other authors. Therefore, in this part of the paper, the results of the present study will be compared with the findings from studies by other authors which investigated the quality of life of children with other rare, neurological or chronic conditions and the impact of those conditions on family functioning.

An analysis of the results of the present study showed that comorbidities have a significant impact on the emotional functioning of MD patients.

The present study also showed that children with MD have a low HRQOL. The parent-reported HRQOL scores of the children studied were lowest for physical functioning and highest for emotional functioning. Similar findings were reported by Rozensztrauch et al. [26] in their study investigating the quality of life of children with Rett syndrome. The study found that the HRQOL scores of the Rett syndrome patients studied were lowest for physical functioning and highest for emotional functioning. Johansen et al. [27] investigated the HRQOL of two hundred and nine (*n* = 209) children with rare conditions (congenital limb deficiency, arthrogryposis multiplex congenita, Marfan’s syndrome, Ehlers–Danlos syndrome, short stature due to skeletal dysplasia, osteogenesis imperfect, and spina bifida/myelomeningocele). An analysis confirmed the hypothesis that children with rare conditions have a low HRQOL. The study showed that Norwegian children with rare conditions had significantly lower parent-reported HRQOL scores compared with healthy Norwegian children. The study showed that the reduction in HRQOL was greatest for the physical functioning domain and was smallest for the emotional functioning domain. MD affects HRQOL in the same way as other rare diseases, as confirmed by the present study.

The present study found that factors such as the age of the child, number of epileptic seizures a week, feeding method (oral feeding or feeding via a PEG), and treatment with copper histidine did not have a significant relationship with the HRQOL scores of the MD patients studied, both with overall and across particular domains, as assessed by the PedsQL Generic Core Scales.

The diagnosis of MD affects most the cognitive functioning of the patient’s family and family relationships. This means that parents of children with MD have most difficulty in, among others, concentrating and remembering things. This is consistent with the findings from a study by Rozensztrauch et al. investigating the impact of Rett syndrome on family functioning [26]. The parents included in the present study scored lowest on the worry domain of the PedsQL Family Impact Module (they often worry about their child’s health) and the daily activities domain of the questionnaire (they frequently have difficulty carrying out daily activities). Epilepsy, which is common in patients with MD, has a similar impact on family functioning, as confirmed by a study by Rozensztrauch et al. [28]. The study showed that the scores reported by parents of children diagnosed with epilepsy were highest for the cognitive functioning and family relationships domains and lowest for the worry and daily activities domains of the PedsQL Family Impact Module. In a study by Ying et al. [29] investigating the functioning of the families of children with cerebral palsy in Malaysia, the scores reported by the caregivers studied were highest for the social functioning, cognitive functioning, and family relationships domains and lowest for the daily activities domain of the PedsQL Family Impact Module. The authors of the study noted that there is increasing evidence that caring for a sick child with special needs has a negative impact on the child’s caregivers (most often parents), causing them to feel sad, overwhelmed, and worried, which also affects their physical health, which in turn has an impact on family functioning [29].

In their study, Ammann-Schnell et al. [30] investigated the impact of rare, chronic neurological disorders—metachromatic leukodystrophy and pontocerebellar hypoplasia type 2—on the families of affected patients. The study showed that parents of children with metachromatic leukodystrophy and pontocerebellar hypoplasia type 2 reported significantly lower levels of family functioning compared to the parents of healthy children.

The present study demonstrated no statistically significant relationship between the duration of disease, which was calculated from the onset of the first symptom, and the HRQOL of children with MD and the functioning of their families. Gallop et al. [31] carried out a literature review on the impact of developmental and epileptic encephalopathies on caregivers. The authors concluded that there is evidence that developmental and epileptic encephalopathies have negative effects on the physical health and daily and social activities of caregivers and that caring for a child with an encephalopathy can affect the caregiver’s productivity, leisure time, and work. The review showed that the factors that influenced the perceived family functioning in the studies analysed included the child’s age, feeding difficulties, sleep problems, and disease severity. A study by Ammann-Schnell et al. [30] carried out among the families of children with metachromatic leukodystrophy and pontocerebellar hypoplasia type 2 showed that the advanced symptoms of the diseases—such as complete immobility and loss of independent eating (the presence of a PEG tube)—had a significant impact on the functioning of the families.

The present study found that the presence of comorbidities has a significant impact on one of the domains of the HRQOL of children with MD, namely, emotional functioning. MD patients without comorbidities are less likely to experience negative emotions such as sadness, fear, and anger and have less difficulty sleeping and feel less worried compared to patients with comorbidities. However, the present study demonstrated no significant relationship between the presence of comorbidities and the overall HRQOL of the children studied, their physical and social functioning, and the functioning of their families. In a study by Rozensztrauch et al. [26], no significant relationship was found between the presence of comorbidities and the quality of life of children with Rett syndrome.

The present study showed that parents of children with MD have moderate caregiver burden. The highest burden is experienced by those parents who, as a result of caregiving duties, do not have the time to pursue their hobbies, who feel socially isolated, who do not have anyone (without medical training) in their close circle who could take care of their child while they are away, and who do not have contact with other families of children with MD. A study by Piran et al. [32] on the caregiving burden of children with chronic conditions showed that the caregivers included in the study had a moderate caregiving burden. A study by Theodore-Oklota et al. [33] on the burden on the caregivers of children with Kabuki syndrome showed that caring for a patient with Kabuki syndrome poses a burden on the caregivers. The caregivers included in the study reported that the burden they experience is due to the fact that caring for a child with Kabuki syndrome is time consuming and affects their ability to work outside the home and that they worry about the child’s health. In their study, Evkaya Acar et al. [34] investigated the caregiver burden of children with spinal muscular atrophy. The largest proportion of the caregivers included in the study reported that they were under a mild/moderate burden.

The parents of children with MD included in the present study reported high perceived social support, as measured by the Multidimensional Scale of Perceived Social Support. In their study, Grzegorczyk et al. [35] assessed the social support received by parents of patients with neurological disorders (epilepsy, cerebral palsy, and West syndrome) using the Berlin Social Support Scales and the authors’ own questionnaire. The study showed that the parents included in the study received a relatively high amount of social support.

## 6. Limitations of the Study

One of the reasons why most of the values obtained and correlations between particular factors were found to be statistically insignificant may be because of the small number of patients included in the study. The study only included 16 patients with MD. This limitation is due to the prevalence of the condition. Moreover, the classical form of MD leads to death in early childhood, which too had an impact on the sample size, as death was one of the study’s exclusion criteria.

Another factor that could have an impact on the results of the study was that the HRQOL of the MD patients studied was assessed using a generic questionnaire. In my opinion and in the opinion of the parents included in the study, the results of the study would be more reliable if a reliable HRQOL assessment questionnaire specific to MD had been used in the study. However, such a questionnaire has yet to be developed.

## 7. Conclusions

The following conclusions can be drawn from the present study; the aim of which was to assess the HRQOL of patients with MD and the impact of the disease on family functioning:

Comorbidities have a significant impact on the emotional functioning of children with MD. However, they do not affect the functioning of the families of MD patients.

Factors such as the age of the child, number of epileptic seizures a week, feeding method (oral feeding or feeding via a PEG tube), and treatment with copper histidine do not have a significant impact on the HRQOL of children with MD.

The duration of disease does not affect the HRQOL of MD patients and the functioning of their families.

MD has a moderate impact on the functioning of the families of affected children.

The parents of children with MD have a moderate caregiver burden.

## Figures and Tables

**Table 1 jcm-12-01769-t001:** Basic descriptive statistics of variables concerning the perceived caregiver burden and perceived social support, and the Shapiro–Wilk test results.

	M	Me	SD	Sk.	Kurt.	Min.	Max.	W	*P*
Caregiver burden	4.88	5.00	1.93	−1.01	1.77	0.00	8.00	0.91	0.128
Perceived social support	66.94	67.50	12.47	−0.21	−0.83	44.00	84.00	0.95	0.527

M—mean; Me—median; SD—standard deviation; Sk.—skewness; Kurt.—kurtosis; Min.—minimum value; Max.—maximum value; W—Shapiro–Wilk test result; *P*—statistical significance.

**Table 2 jcm-12-01769-t002:** HRQOL of children with MD, including the results of the Shapiro–Wilk test.

	M	Me	SD	Sk.	Kurt.	Min.	Max.	W	*P*
HRQOL—children	29.14	27.40	14.73	0.57	0.19	6.67	61.04	0.97	0.767
Physical functioning—children	10.55	7.82	10.26	0.88	−0.29	0.00	31.25	0.88	0.034
Emotional functioning—children	48.13	47.50	29.43	−0.35	−0.91	0.00	90.00	0.93	0.213
Social functioning—children	28.75	25.00	18.57	0.69	1.46	0.00	75.00	0.93	0.266

M—mean; Me—median; SD—standard deviation; Sk.—skewness; Kurt.—kurtosis; Min.—minimum value; Max.—maximum value; W—Shapiro–Wilk test result; *P*—statistical significance.

**Table 3 jcm-12-01769-t003:** HRQOL in MD and control group.

PedsQL Score	*n*	Me	SD	*n*	Me	SD
		Study group			Control group	
Total score	16	27.40	14.73	6530	80.40	16.10
Physical functioning	16	7.82	10.26	6519	82.11	20.63
Emotional functioning	16	47.50	29.43	6517	80,00	17.25
Social functioning	16	25.00	18.57	6533	79,49	15.87

M—mean; SD—standard deviation.

**Table 4 jcm-12-01769-t004:** Impact of MD on family functioning, including the results of the Shapiro–Wilk test.

	M	Me	SD	Sk.	Kurt.	Min.	Max.	W	P
Family functioning—parents	44.16	42.50	17.50	0.07	−0.06	13.23	78.13	0.97	0.891
Physical functioning—parents	39.84	39.59	14.90	−0.82	0.62	4.17	58.33	0.93	0.279
Emotional functioning—parents	45.00	50.00	20.17	−0.09	0.01	5.00	85.00	0.98	0.974
Social functioning—parents	41.80	37.50	24.23	0.58	−0.21	6.25	87.50	0.94	0.393
Cognitive functioning—parents	50.00	52.50	19.24	0.12	−0.07	20.00	90.00	0.94	0.366
Communication	48.44	41.67	22.61	0.23	−0.25	8.33	91.67	0.96	0.717
Worry	35.31	32.50	18.93	0.36	−0.55	5.00	70.00	0.96	0.689
Family functioning summary score	44.27	44.17	17.49	0.05	−1.08	14.17	71.67	0.96	0.669
Daily activities	32.29	33.33	20.38	−0.27	−1.15	0.00	58.33	0.92	0.157
Family relationships	56.25	55.00	22.10	−0.02	−0.74	20.00	90.00	0.94	0.392

M—mean; Me—median; SD—standard deviation; Sk.—skewness; Kurt.—kurtosis; Min.—minimum value; Max.—maximum value; W—Shapiro–Wilk test result; *P*—statistical significance.

**Table 5 jcm-12-01769-t005:** Correlation between age and the number of epileptic seizures a week and the HRQOL of the children studied.

		Overall HRQOL—Children	Physical Functioning	Emotional Functioning	Social Functioning
Age	Pearson’s r	0.34	0.17	0.27	0.29
	significance	0.193	0.519	0.315	0.268
Number of epileptic seizures a week	Spearman’s rho	−0.13	−0.04	−0.12	−0.07
	significance	0.641	0.891	0.663	0.801

**Table 6 jcm-12-01769-t006:** Comparison of the HRQOL of the groups of children studied differing in terms of the feeding method and in terms of whether they are treated with copper histidine.

Dependent Variable	Independent Variable		M	SD	T	p
Total HRQOL score—children	Feeding method	Oral feeding (*n* = 7)	35.66	16.61	1.29	0.222
		PEG (*n* = 7)	25.95	10.97		
	Treatment with copper histidine	No (*n* = 7)	29.61	14.08	0.11	0.914
		Yes (*n* = 9)	28.77	16.05		
Physical functioning	Feeding method	Oral feeding (*n* = 7)	13.39	13.10	0.48	0.642
		PEG (*n* = 7)	10.72	6.72		
	Treatment with copper histidine	No (*n* = 7)	10.27	12.85	−0.09	0.927
		Yes (*n* = 9)	10.77	8.57		
Emotional functioning	Feeding method	Oral feeding (*n* = 7)	54.29	31.94	0.51	0.622
		PEG (*n* = 7)	46.43	25.77		
	Treatment with copper histidine	No (*n* = 7)	51.43	36.25	0.38	0.706
		Yes (*n* = 9)	45.56	24.93		
Social functioning	Feeding method	Oral feeding (*n* = 7)	39.29	19.67	1.95	0.074
		PEG (*n* = 7)	20.71	15.66		
	Treatment with copper histidine	No (*n* = 7)	27.14	9.51	−0.33	0.751
		Yes (*n* = 9)	30.00	23.98		

M—mean; SD—standard deviation; t—Student’s *t*-test statistics; p—statistical significance.

**Table 7 jcm-12-01769-t007:** Correlation between duration of disease and the HRQOL of children with MD and the functioning of their families.

		Duration of Disease
Total HRQOL score—children	Pearson’s r significance	0.34 0.199
Physical functioning	Pearson’s r significance	0.18 0.497
Emotional functioning	Pearson’s r significance	0.26 0.329
Social functioning	Pearson’s r significance	0.29 0.272
Family functioning summary score	Pearson’s r significance	−0.16 0.560
Daily activities	Pearson’s r significance	−0.07 0.787
Family relationships	Pearson’s r significance	−0.18 0.501

**Table 8 jcm-12-01769-t008:** Comparison of children with and without comorbidities in terms of their HRQOL and the functioning of their families.

Dependent Variable	Comorbidities	M	SD	T	*P*
Total HRQOL score—children	No (*n* = 8)	35.60	16.10	1.90	0.078
	Yes (*n* = 8)	22.68	10.49		
Physical functioning	No (*n* = 8)	10.55	12.61	0.00	1.000
	Yes (*n* = 8)	10.55	8.17		
Emotional functioning	No (*n* = 8)	63.75	22.80	2.45	0.028
	Yes (*n* = 8)	32.50	27.90		
Social functioning	No (*n* = 8)	32.50	22.68	0.80	0.438
	Yes (*n* = 8)	25.00	13.89		
Family functioning summary score	No (*n* = 8)	41.98	15.84	−0.51	0.617
	Yes (*n* = 8)	46.56	19.81		
Daily activities	No (*n* = 8)	27.08	22.60	−1.02	0.323
	Yes (*n* = 8)	37.50	17.82		
Family relationships	No (*n* = 8)	56.88	22.67	0.11	0.914
	Yes (*n* = 8)	55.63	23.06		

M—mean; SD—standard deviation; t—Student’s *t*-test statistics; *P*—statistical significance.

## Data Availability

The data that support the findings of this study are available from the corresponding author, upon reasonable request.

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
