# Peer review of "Health-Related Quality of Life and Family Functioning of Primary Caregivers of Children with Menkes Disease"

_jcm, 2023, doi:10.3390/jcm12051769_

Round 1

Reviewer 1 Report

In the present article, submitted to JCM, Rozensztrauch and colleagues investigate the Quality of Life of children with a rare Mendelian disorder of copper metabolism named Menke´s disease and their caregivers´ assessement of family functioning.

I think the study is very interesting and from my point of view it was carried out with the required consistency and complexity, I have some minor comments that the authors could still address to make the study more interesting for a wider readership.

Major points:

1. Paragraph 3 (Statistical analysis) contains quite a few repetitive information and should be re-written in my view.

Example: Lines 229-234: “The next step of the analysis was to assess whether the child’s age, number of epileptic seizures a week, feeding method and treatment with copper histidine have a relationship with the HRQOL of children with MD…” contains information very similar to lines 235-239.

2. In general, one limitation is that the values are difficult to relate to control cohorts (e.g., of another neurodegenerative disease or healthy children), among other things because studies of different investigators have different core questions and thus an inter-study standardization is difficult - even when using partly the same questionnaires. However, the statement: "MD has a moderate impact on the functioning of families of affected children." I find difficult especially under the aspect of the small cohort. In my opinion, questions about the "before" are missing, i.e. questions about the social life of the families before the symptom onset of the affected children. Or is this statement also to be understood in the context of other severe neurological diseases. Apparently Menke's disease is no exception, because the authors came to a similar conclusion in their study on Rett syndrome.

Minor points:

1. In Material and Methods the first sentence reads:” The youngest child was 5 months old and the oldest was 15 years old. Nearly half of the children were under the 151 age of 3.” In results, the first sentence repeats this information. Although I think that in principle this describes the cohort sufficiently, a tabulation of individuals or cohort statistics would be a better alternative. I personally see no specific limitation in that most children were aged under 3. But a pediatrician or Menke specialist might be interested in the age distribution of children between 3 - 15 years.

2. The authors themselves point out the main limitation of the small and heterogeneous cohort. In other studies, parents of healthy children are still used as a control cohort. I think it's reasonable not to have done that here, but couldn't the authors put these data here maybe from the literature in the corresponding analysis tables. From my point of view, it would be at most Table 1 and Table 2.

3. Is the high score for the emotional functioning domain explainable or is it only high compared to the physical functioning domain score and actually also rather low? Or do the caregivers tend to compensate for the physical limitations of the children?

4. Instead of or in addition to the analysis depending on the HRQOL of comorbidities (Table 7), an analysis depending on whether the child is in hospice care could be performed.

Hospice care should have a massive impact on daily family life (social functioning)! Or is it not a 24/7 hospice care? Maybe the authors can more explain here possible differences in the living situation of the 16 participating families if they have knowledge about it.

Author Response

Dear Reviewer 1,

Thank you very much for sending us the consensus opinion about requested revision of our manuscript entitled: Health-Related Quality of Life and Family Functioning of Primary Caregivers of Children with Menkes Disease. We appreciate the thoughtful comments and we have modified the manuscript in response to your suggestions, which we believe will further improve its quality. 

REVIEWER COMMENTS 1

In the present article, submitted to JCM, Rozensztrauch and colleagues investigate the Quality of Life of children with a rare Mendelian disorder of copper metabolism named Menke´s disease and their caregivers´ assessement of family functioning.

I think the study is very interesting and from my point of view it was carried out with the required consistency and complexity, I have some minor comments that the authors could still address to make the study more interesting for a wider readership.

Thank you for this comment.

REVIEWER COMMENTS 2

Major points:

Paragraph 3 (Statistical analysis) contains quite a few repetitive information and should be re-written in my view.

Example: Lines 229-234: “The next step of the analysis was to assess whether the child’s age, number of epileptic seizures a week, feeding method and treatment with copper histidine have a relationship with the HRQOL of children with MD…” contains information very similar to lines 235-239.

Thank you for this comment. After the reviewer suggestion we crossed out the entire sentence lines 235 – 239.

REVIEWER COMMENTS 2

In general, one limitation is that the values are difficult to relate to control cohorts (e.g., of another neurodegenerative disease or healthy children), among other things because studies of different investigators have different core questions and thus an inter-study standardization is difficult - even when using partly the same questionnaires. However, the statement: "MD has a moderate impact on the functioning of families of affected children." I find difficult especially under the aspect of the small cohort. In my opinion, questions about the "before" are missing, i.e. questions about the social life of the families before the symptom onset of the affected children. Or is this statement also to be understood in the context of other severe neurological diseases. Apparently Menke's disease is no exception, because the authors came to a similar conclusion in their study on Rett syndrome.

            Thank you for this comment. I  deeply agree that the question about “before” is a very good suggestion. Thank you very much for the suggestions, the authors will take it into account in future research.

REVIEWER COMMENTS 3

Minor points:

In Material and Methods the first sentence reads:” The youngest child was 5 months old and the oldest was 15 years old. Nearly half of the children were under the 151 age of 3.” In results, the first sentence repeats this information. Although I think that in principle this describes the cohort sufficiently, a tabulation of individuals or cohort statistics would be a better alternative. I personally see no specific limitation in that most children were aged under 3. But a pediatrician or Menke specialist might be interested in the age distribution of children between 3 - 15 years.

Thank you for this comment . After reviewer suggestion we have added the following sentences.

…”Almost half of the children (44%) were aged under 3, between 3 and 6 years old were 19%, between 6 and 10 years old were 6%, over 10 years old were 31%...”

REVIEWER COMMENTS 4

The authors themselves point out the main limitation of the small and heterogeneous cohort. In other studies, parents of healthy children are still used as a control cohort. I think it's reasonable not to have done that here, but couldn't the authors put these data here maybe from the literature in the corresponding analysis tables. From my point of view, it would be at most Table 1 and Table 2.

Thank you for this comment. After reviewer suggestion we have incorporated new table 3 in the results section.

PedsQL

score

n

Me

SD

n

Me

SD

Study group

Control group [23]

Total score

16

27.40

14.73

6530

80.40

16.10

Physical functioning

16

7.82

10.26

6519

82.11

20.63

Emotional

functioning

16

47.50

29.43

6517

80,00

17.25

Social functioning-children

16

25.00

18.57

6533

79,49

15.87

Table 3. HRQOL in MD and control group.

REVIEWER COMMENTS 5

Is the high score for the emotional functioning domain explainable or is it only high compared to the physical functioning domain score and actually also rather low? Or do the caregivers tend to compensate for the physical limitations of the children?

Thank you for this comment.

REVIEWER COMMENTS 6

Instead of or in addition to the analysis depending on the HRQOL of comorbidities (Table 7), an analysis depending on whether the child is in hospice care could be performed. Hospice care should have a massive impact on daily family life (social functioning)! Or is it not a 24/7 hospice care? Maybe the authors can more explain here possible differences in the living situation of the 16 participating families if they have knowledge about it.

Thank you for this comment. Unfortunately, we did not investigate hospice care on daily family life.

Reviewer 2 Report

This study deals with a rare disease in copper metabolism. In doing so, she investigates the social burden on families with children of Menke disease.

Menke's disease is presented in great detail in the introduction.

The results are in line with expectations. They prove the burden of a family with a child with a severe disability. This is not new and is also known in other diseases.

The discussion is very long-winded and does not bring any significant new insights into the disease of Menke's disease.

Nevertheless, in view of the rarity of the disease, it seems to me worth mentioning a communication of the family social burden. For this, the contribution should be significantly reduced.

Author Response

Dear Reviewer 2,

Thank you very much for sending us the consensus opinion about requested revision of our manuscript entitled: Health-Related Quality of Life and Family Functioning of Primary Caregivers of Children with Menkes Disease. We appreciate the thoughtful comments and we have modified the manuscript in response to your suggestions, which we believe will further improve its quality. 

REVIEWER COMMENTS 1

This study deals with a rare disease in copper metabolism. In doing so, she investigates the social burden on families with children of Menke disease.

Menke's disease is presented in great detail in the introduction.

Thank you for this comment.

REVIEWER COMMENTS 2

The results are in line with expectations. They prove the burden of a family with a child with a severe disability. This is not new and is also known in other diseases.

Thank you for this comment.

REVIEWER COMMENTS 3

The discussion is very long-winded and does not bring any significant new insights into the disease of Menke's disease. Nevertheless, in view of the rarity of the disease, it seems to me worth mentioning a communication of the family social burden. For this, the contribution should be significantly reduced.

Thank you for this comment. Yes, indeed the discussion for some readers may be too long but from our point of view it is needed in this form due to a rare disease.
